# Transductive Universal Transport for Zero-Shot Action Recognition

## Abstract

This work addresses the problem of recognizing action categories in videos for which no training examples are available. The current state-of-the-art enables such a zero-shot recognition by learning universal mappings from videos to a shared semantic space, either trained on large-scale seen actions or on objects. While effective, we find that universal action and object models are biased to their seen categories. Such biases are further amplified due to biases between seen and unseen categories in the semantic space. The amplified biases result in many unseen action categories simply never being selected during inference, hampering zero-shot progress. We seeks to address this limitation and introduce transductive universal transport for zero-shot action recognition. Our proposal is to re-position unseen action embeddings through transduction, *i.e.,* by using the distribution of the unlabelled test set. For universal action models, we first find an optimal mapping from unseen actions to the mapped test videos in the shared hyperspherical space. We then propose new target embeddings as weighted Fréchet means, with the weights given by the transport couplings. Finally, we introduce a re-positioning of unseen action embeddings along the geodesic between the original and target, as a form of semantic regularization. For universal object models, we outline a weighted transport variant from unseen action embeddings to object embeddings directly. Empirically, we show that our approach directly boosts universal action and object models, resulting in state-of-the-art performance for zero-shot classification and spatio-temporal localization.

## 1 Introduction

This paper investigates the problem of inferring actions in videos for which no training examples were provided. Akin to advances in the image domain (He et al., 2016; Huang et al., 2017; Yuan et al., 2021), deeper and more expressive network architectures have in recent years been developed to continually improve the recognition of actions in videos (Carreira & Zisserman, 2017; Feichtenhofer et al., 2019; Arnab et al., 2021). To meet the corresponding demand for data, multiple new datasets – each spanning many hundreds of actions – have been created, *e.g.,* (Carreira & Zisserman, 2017; Monfort et al., 2019; Damen et al., 2020). While such datasets increase the coverage of the action space, we seek to recognize actions even when no examples are available during training.

In zero-shot action recognition, one line of work has found success by mirroring progress in its image-based counterpart, for example by using attributes (Liu et al., 2011; Gan et al., 2016b) or feature synthesis (Mishra et al., 2020) to transfer knowledge from seen to unseen actions. Recently, state-of-the-art results have been achieved by taking a universal learning perspective, where large-scale models are trained to map input videos to a shared semantic space occupied by both seen and unseen categories. In the first universal perspective, large-scale networks train a mapping from videos to a semantic space on hundreds of seen actions from *e.g.,* ActivityNet (Zhu et al., 2018) or Kinetics (Brattoli et al., 2020). For a target dataset with unseen actions, zero-shot inference is directly possible through a nearest neighbour search between the video mappings and the embeddings of unseen actions. In the second perspective, networks are trained on thousands of objects (Jain et al., 2015; Mettes et al., 2021), where inference becomes a function of both the object likelihoods and the semantic relation between objects and actions.

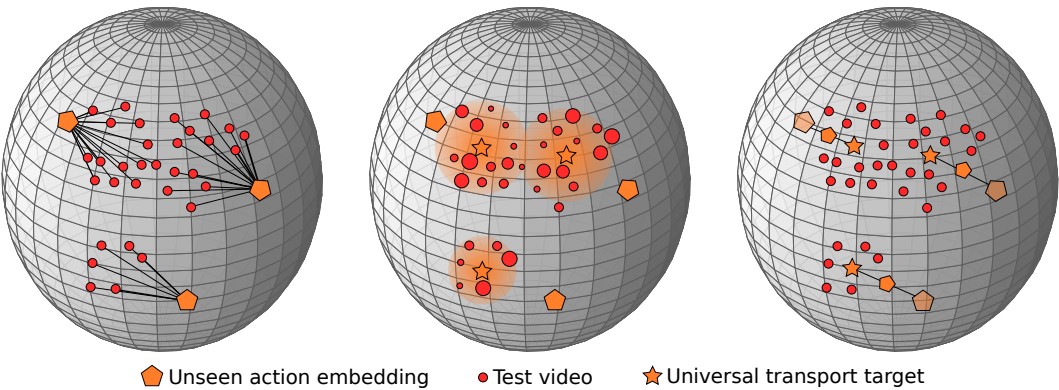

Unseen action embedding · Test video ★ Universal transport target

Figure 1: **Overview of transductive universal transport** for universal action models. First, we find an optimal mapping from the discrete measure representing unseen actions to the discrete measure representing the mapped test videos. Second, we define the target embedding for each unseen action as Fréchet means over the test videos, weighted by their optimal coupling. Third, we re-position unseen action embeddings along the geodesic spanned by the original and target embeddings.

While effective, the first finding of this work is that both perspectives share a common limitation: they are strongly biased to subsets of the unseen actions. Compounding biases from videos to seen categories and from seen to unseen categories result in a mismatch between the embeddings of unseen actions and mapping of the videos covering the same actions. A significant part of the actions are as a result simply never selected, disrupting progress in zero-shot action recognition. We seek to address this limitation.

In this work, we introduce transductive universal transport for zero-shot action recognition with universal models. The idea is to re-position the semantic embeddings of unseen actions by using information about the distribution of the entire test set. For universal action models, we first consider the optimal transport mapping between the semantic embeddings of unseen actions and all embedded videos. We then propose a target function for each unseen action as a weighted Fréchet mean given the optimal transport couplings. To avoid loosing semantic interpretation, we finally propose to re-position unseen actions along the geodesic spanned by the original and target embeddings. We view this as a form of semantic regularization imposed on the target embeddings. For universal object models, we perform the same approach with two differences: we map the unseen actions to object embeddings directly and we add transport weights to both the unseen actions and the objects, such that we focus on actions far away from objects and on objects that are more likely to occur in the test videos. Experiments on four datasets validate our approach, with state-of-the-art results for zero-shot action recognition and spatio-temporal action localization.

## 2 RE-POSITIONING ACTION EMBEDDINGS WITH UNIVERSAL TRANSPORT

For the problem of zero-shot action recognition, we are given a set of test videos $\mathcal{V}_u$ and a set of labels $\mathcal{L}_u$ denoting actions which have not been observed during training. We seek to assign a label $l \in \mathcal{L}_u$ to each test video. To that end, we start from two state-of-the-art universal learning directions in zero-shot action recognition, namely by transferring knowledge from large-scale seen actions and from objects. For both directions, we seek to overcome the long-tailed distributions in label assignment of unseen actions during zero-shot inference. Below, we first introduce transductive universal transport for the transfer from seen to unseen actions, see Figure 1. Second, we extend our approach for re-positioning unseen action embeddings based on objects.

### 2.1 TRANSDUCTIVE UNIVERSAL TRANSPORT FOR UNIVERSAL ACTION MODELS

Inferring unseen actions by transferring knowledge from seen actions requires two transformation functions: a function $\omega$ that maps a label to a semantic embedding space and a function $\phi$ that maps a video to the same embedding space. The mapping function $\phi$ is learned on a set of training videos

$\mathcal{V}_s$ with seen action labels $\mathcal{L}_s$, with its loss given as:

$$\mathcal{L}_s = \sum_{v \in \mathcal{V}_s} -\frac{\langle \phi(v), \omega(l_v) \rangle}{||\phi(v)|| \cdot ||\omega(l_v)||}, \tag{1}$$

with $l_v \in \mathcal{L}_s$ the action label for video $v$. The function $\omega$ is given by a pre-trained word embedding model (Mikolov et al., 2013b), where embeddings are commonly $\ell_2$-normalized and optimized with the cosine distance. Due to the shared semantic nature of the embedding space, any test video that is projected to the embedding space using $\phi$ can also be assigned to a label from $\mathcal{L}_u$ through a nearest neighbour search with all unseen labels that have been mapped using $\omega$. In this work, we argue that since $\phi$ is trained on actions not used during inference, the mapped video embeddings and unseen action embeddings are not aligned. We propose to improve zero-shot action recognition by re-positioning unseen action embeddings with transductive information.

In order to better position unseen action embeddings, we first redefine the original seen and unseen action embeddings as discrete measures so that they can be used in the context of optimal transport:

**Definition 1** (Actions as discrete measures). *The sets of seen and unseen action labels are given as:*

$$\mu_s = \sum_{l_s \in \mathcal{L}_s} w_{l_s} \delta_{\omega(l_s)}, \qquad \mu_u = \sum_{l_u \in \mathcal{L}_u} w_{l_u} \delta_{\omega(l_u)}, \tag{2}$$

*where $w_{l_u} \in \mathbf{w}_u$ and $w_{l_s} \in \mathbf{w}_s$ denote the set of weights for the action, such that $\mathbf{w}_s \in \Sigma_{|\mathcal{L}_s|}$ and $\mathbf{w}_u \in \Sigma_{|\mathcal{L}_u|}$ are both on the probability simplex and $\delta_x$ denotes the Dirac at position $x$.*

In similar fashion the definition of the mapped test videos is given as:

**Definition 2** (Videos as a discrete measure). *The set of mapped test videos is given as:*

$$\mu_v = \sum_{c \in C^k(\mathcal{V}_u)} w_c \delta_c, \qquad c = \frac{1}{|a(\mathcal{V}_u; c)|} \sum_{v \in a(\mathcal{V}_u; c)} \phi(v), \tag{3}$$

*with $w_c \in \mathbf{w}_c \in \Sigma_k$ akin to Definition 1, where $C^k$ denotes a k-component cluster aggregation over the set of videos, and where $a(\mathcal{V}_u; c)$ denotes the set of videos assigned to cluster c.*

With the above definitions, seen and unseen actions are interpreted as discrete measures centered around their word embeddings. For the test videos, we define the corresponding discrete measure as a clustering over the mapped video outputs. This is to make the measure itself more robust to outliers and to increase the focus towards high-density regions. The seen actions are only used to train the universal network $\phi(\cdot)$; the unseen actions and test videos are central in our approach.

With the unseen action labels and videos defined as discrete measures in the same space, we first seek to find an optimal transport mapping between the two measures. Since we operate on the entire distribution of test videos rather than performing inference for each video independently, we view this as a transductive form of optimal transport, see Figure 1 (left). Although the action embeddings are discrete entities, we find that it is better to view the transport as an instance of the relaxed Monge-Kantorovich problem, which allows for mass splitting from the unseen action measure to the aggregated video measure. Let $\mathbf{C}$ denote the cost matrix, with $C_{ij}$ defined as the cosine distance between action $i$ and video cluster $j$. Then the transductive optimal transport form $\mu_u$ to $\mu_v$ is given as (Peyré et al., 2019):

$$\mathcal{L}_{\mathbf{C}}(\mathbf{w}_u, \mathbf{w}_v) = \min_{\mathbf{P} \in \mathbf{U}(\mathbf{w}_u, \mathbf{w}_v)} \langle \mathbf{C}, \mathbf{P} \rangle = \sum_{ij} \mathbf{C}_{ij} \mathbf{P}_{ij}, \tag{4}$$

where $\mathbf{U}$ denotes the admissible couplings, given as:

$$\mathbf{U}(\mathbf{w}_u, \mathbf{w}_v) = \left\{ \mathbf{P} \in \mathbb{R}_+^{|\mathcal{L}_u| k} : \mathbf{P}\mathbf{1}_k = \mathbf{w}_u \, \& \, \mathbf{P}\mathbf{1}_{|\mathcal{L}_u|} = \mathbf{w}_v \right\},$$

$$\mathbf{P}\mathbf{1}_k = (\sum_j \mathbf{P}_{ij})_i, \quad \mathbf{P}\mathbf{1}_{|\mathcal{L}_u|} = (\sum_i \mathbf{P}_{ij})_j, \tag{5}$$

which is solved using the Lagrangian approach of Bonneel et al. (2011). This results in a coupling matrix $\mathbf{P}$, where $\mathbf{P}_{ij}$ denotes the amount of mass of action $i$ that moves to video cluster $j$. When working with universal action models, we set the weights of both measures uniformly.

Given the optimal transport mapping, we propose to condense the corresponding coupling into a single target embedding per unseen action. Since since the semantic space on which we operate is a hypersphere, we define the target embedding as the weighted Fréchet mean (Lou et al., 2020; Miolane et al., 2020) using normalized coupling values:

$$\omega^{\text{target}}(l_i) = \arg\min_{s \in \mathbb{S}^{d-1}} \sum_{j=1}^{k} \hat{\mathbf{P}}_{ij} d(c_j, s)^2, \qquad \sum_{j=1}^{k} \hat{\mathbf{P}}_{ij} = 1, \qquad (6)$$

with $d$ the cosine similarity, see Figure 1 (middle). The target provides a new location in embedding space for each unseen action label, guided by the distribution of mapped and clustered test videos. While we can directly use the new embeddings for inference, we want to avoid big changes in embedding space since that relates with losing the original semantic interpretation of the action. We therefore dictate that the final embedding of each unseen action is positioned along the geodesic spanned by the original and target embeddings, modelled through spherical interpolation:

$$\omega^{\star}(l) = \frac{\sin[\lambda\Omega]}{\sin\Omega}\omega(l) + \frac{\sin[(1-\lambda)\Omega]}{\sin\Omega}\omega^{\text{target}}(l), \qquad \cos\Omega = \langle\omega(l), \omega^{\text{target}}(l)\rangle, \qquad (7)$$

where $\lambda$ denotes the balance between maintaining the original embedding and moving towards the transductive target embedding. In this manner, we move each unseen action towards its proposed target embedding, with the interpolation acting as a regularization towards the original embedding. Zero-shot action recognition is performed by computing the cosine similarity between the projected video $v$ and each unseen action label $l$ as

$$s_{\text{action}}(l|v) = \frac{\langle\phi(v), \omega^{\star}(l)\rangle}{||\phi(v)|| \cdot ||\omega^{\star}(l)||}, \qquad (8)$$

after which the action label with the highest similarity is selected. Figure 1 (right) provides an illustration of the final re-positioning, where we move the original unseen action embeddings along the greater arc of the hypersphere towards the optimal target embeddings.

## 2.2 TRANSDUCTIVE UNIVERSAL TRANSPORT FOR UNIVERSAL OBJECT MODELS

Universal object models form a competitive and similarly biased alternative for zero-shot action recognition. To tackle these biases, we also redefine objects as discrete measures to make them suitable in the context of optimal transport:

**Definition 3** (Objects as a discrete measure). *The sets of object labels are given as:*

$$\mu_o = \sum_{o \in \mathcal{O}_s} w_o \delta_{\omega(o)}, \qquad \mathcal{O}_s = \{o \in \mathcal{O} \mid \max_{v \in \mathcal{V}_u} p(o|v) \geq \tau\}, \qquad (9)$$

*with $w_o \in \mathbf{o}_c$ and where $p(o|v)$ denotes the likelihood of object $o$ occurring in video $v$.*

Different from both the action and video measure definitions, the discrete measure for objects is based on a subset $\mathcal{O}_s \in \mathcal{O}$, *i.e.,* we define a degenerate distribution over the objects. The subset is determined by again taking a transductive view; we exclude any object which does not have a likelihood over a low threshold $\tau$ in *any* test video. The idea behind this definition is to avoid a bias in the optimal transport to objects which do not actually occur in videos.

Akin to our approach for universal action models, we compute a coupling between the unseen action measure and the object measure, as given by Equations 4 and 5. Differently however, we set non-uniform weights for both the actions and objects. The objects are weighted according to their transductive maximum score, $w_o = \max_{v \in \mathcal{V}_u} p(o|v)/\mathcal{Z}_o$, with $\mathcal{Z}_o$ a normalization constant over all objects in $\mathcal{O}_s$. The unseen action weights are given as $w_a = (1 - ((\max_{o \in \mathcal{O}_s}\langle\omega(a), \omega(o)\rangle/2) + 0.5))/\mathcal{Z}_a$, with $\mathcal{Z}_a$ a normalization constant over all actions. The intuition behind the object weights is to focus the attention of the transductive universal transport on objects with a higher visual likelihood. The weights for the unseen actions are given as the inverse over the maximum word embedding similarity with respect to the objects, under the notion that actions without obvious relations to objects should have a more prominent spot in the transport coupling. With the optimal transport computed between unseen actions and objects, action embeddings are again interpolated following Equations 6 and 7, with the updated embedding for action label $l$ now denoted as $\omega^{\ddagger}(l)$.

For the final zero-shot action inference from objects, we follow the same setup as current object-based approaches, where the score for each action label $l$ in video $v$ is determined based on the top relevant objects for that action (Jain et al., 2015; Mettes et al., 2021):

$$s_{\text{object}}(l|v) = \sum_{o \in \mathcal{O}_l} p(o|v) \cdot \frac{\langle \omega(o), \omega^{\ddagger}(l) \rangle}{||\omega(o)|| \cdot ||\omega^{\ddagger}(l)||}. \tag{10}$$

with $\mathcal{O}_l$ the set of most semantically similar objects for action label $l$. Finally, the transductive action-based and object-based scores from respectively Equations 8 and 10 can also be fused as $s_{\text{fusion}}(l|v) = \epsilon \cdot s_{\text{action}}(l|v) + (1 - \epsilon) \cdot s_{\text{object}}(l|v)$.

## 3 EXPERIMENTAL SETUP

### 3.1 DATASETS

**Source datasets.** We employ two source datasets for universal representation learning, namely Kinetics-700 for actions and ImageNet for objects. For **Kinetics-700**, we follow Brattoli et al. (2020) and use a subset with 664 action categories to avoid any overlap with action categories in datasets used for zero-shot action recognition. For **ImageNet**, we follow Mettes et al. (2021) and use the reorganized variant containing 12,988 object categories (Mettes et al., 2020).

**Target datasets.** The classification evaluation is performed on the two datasets used most often in zero-shot action recognition: UCF-101 and HMDB51. The **UCF-101** dataset consists of 13,320 videos covering 101 action categories. Next to 101-way zero-shot evaluation, we also evaluate on settings with 20 and 50 test actions. For these settings, we rerun our approach on 10 runs with randomly selected actions and we report the mean and standard deviation over the runs. We note that in the 20- and 50-way zero-shot recognition, we do not use the other actions for training, they are simply not used in our approach. The **HMDB51** dataset consists of 6,766 videos covering 51 action categories. Next to 51-way evaluation, we also investigate 10- and 25-way zero-shot recognition.

We also investigate the potential of our approach for zero-shot spatio-temporal action localization on UCF Sports and J-HMDB. **UCF Sports** consists of 150 videos with 10 actions and **J-HMDB** consists of 928 videos with 21 actions. For the evaluation, we follow Jain et al. (2015) and report results with the AUC metric across five overlap thresholds.

### 3.2 IMPLEMENTATION DETAILS

For the universal action network $\phi$, we employ the R(2+1)D network (Tran et al., 2018) as given by Brattoli et al. (2020), which is pre-trained on the 664 action categories from Kinetics. For each video, we obtain its corresponding video embedding by randomly selecting a 16 frame shot and passing the shot through $\phi$. For a fair comparison to the state-of-the-art by Brattoli et al. (2020), we also use the same word embedding $\omega$, namely a word2vec model (Mikolov et al., 2013a), resulting in a 300-dimensional representation per word. For any action or object with more than one word, the word representations are averaged. The object scores in a video are obtained following the protocol of Mettes et al. (2021), where two frames per second are sampled, each fed to the pre-trained ImageNet model, and with the object probabilities averaged over the sampled frames. For the clustering of the video embeddings in Definition 2, we use k-means clustering along with $\ell_2$-normalization, akin to Banerjee et al. (2005). For the optimal transport, we employ the Lagrangian approach of Bonneel et al. (2011) as implemented in (Flamary et al., 2021). Lastly for spatio-temporal localization, we start from the tubes made available by Mettes et al. (2021). To each tube, we add the corresponding video-level action scores from our approach to improve the ranking of the action tubes over the entire dataset. All code will be made publicly available.

## 4 EXPERIMENTAL RESULTS

We focus on three experiments: (i) evaluations on universal action models, (ii) evaluations on universal object models along with the fusion between both, and (iii) state-of-the-art comparison for zero-shot action recognition and zero-shot spatio-temporal action localization.

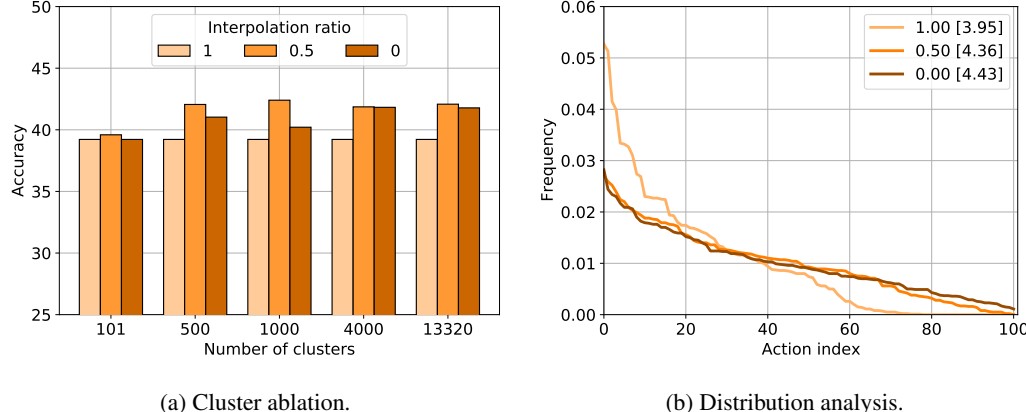

(a) Cluster ablation.

(b) Distribution analysis.

Figure 2: **Evaluating transductive universal transport from seen actions** on UCF-101. Left: The effect of the number of clusters and the interpolation ratio on the recognition performance, showing the positive effect of re-positioning unseen actions and the semantic regularization given by the spherical interpolation. Right: An intuition behind the obtained results. Using the original unseen action embeddings results in large biases during zero-shot inference, here using 1,000 clusters. With out approach, this imbalance is reduced, as indicated by the more even action distributions in the plot and the corresponding higher entropy scores in brackets in the legend.

## 4.1 EVALUATING TRANSDUCTIVE UNIVERSAL TRANSPORT FROM SEEN ACTIONS

**Setup.** For the first experiment, we evaluate on UCF-101 using all 101 actions for classification. We investigate the two variables that come with our approach in the context of universal action models, namely the granularity of the cluster aggregation over all test videos and the interpolation ratio between the original and target embeddings of the unseen actions.

**Results.** The results for six cluster sizes and three interpolation ratios are shown in Figure 2a. An interpolation rate of 1 denotes only using the original action embeddings and 0 only the target embeddings. With the original embeddings, we obtain an accuracy of 39.2%. Using the target embeddings from the universal transport ($\lambda = 0$) directly boosts the classification accuracy when using many clusters, while results degrade for few clusters. The best performance is obtained by positioning the unseen actions halfway along the geodesic between the original and target embeddings. With 1,000 clusters the accuracy becomes 42.4%, bcompared to 40.1% without interpolation.

**Analysis.** An explanation for our improvements is shown in Figure 2b. We show the distributions of selected actions across all interpolation ratios when using 1,000 clusters. With the original unseen action embeddings, this distribution is highly uneven, with 23% (!) of the actions never being selected, naturally leading to zero accuracy for these actions. With our approach, the distributions become more uniform, highlighting the deduced bias present in our approach. We conclude that transductive universal transport improves zero-shot action recognition for universal action models. Throughout the rest of the experiments, we use 1,000 clusters with an interpolation ratio of 0.5.

## 4.2 EVALUATING TRANSDUCTIVE UNIVERSAL TRANSPORT FROM OBJECTS

**Setup.** Next, we investigate our approach on universal object models. We again use UCF-101 with all 101 actions for evaluation, with the interpolation ratio fixed to 0.5. We evaluate three threshold levels that come with the definition of objects as discrete measure, along with the universal object model itself and a vanilla uniformly-weighted optimal transport using all objects as baselines.

**Results.** In Table 1 on the left, we show the zero-shot action results for our approach when maintaining the top 2,500, 1,000, and 500 objects according to their transductive maximum probability over all test videos. We first find that using a baseline optimal transport approach akin to the setup for seen actions provides only a marginal boost from 29.2% to 30.1%. In contrast, using the proposed weights for the unseen actions and objects, combined with a filtering of objects never present in a

| UCF-101 | | |
|---|---|---|
| | $|\mathcal{O}_s|$ | Accuracy |
| Baseline | - | 29.9 |
| Baseline transport | 12,988 | 30.1 |
| Proposed transport | 2,500 | 31.4 |
| Proposed transport | 1,000 | **31.6** |
| Proposed transport | 500 | 30.2 |

| UCF-101 | | |
|---|---|---|
| Weighting | | Accuracy |
| Actions | Objects | |
| uniform | uniform | 29.8 |
| inverse | uniform | 30.4 |
| uniform | transductive | 30.6 |
| inverse | transductive | **31.6** |

Table 1: **Evaluating transductive universal transport from objects** on UCF-101. Our proposed approach also enhances universal object-based approaches for zero-shot action recognition, due to both object filtering (left) and the proposed weighting of unseen actions and objects (right).

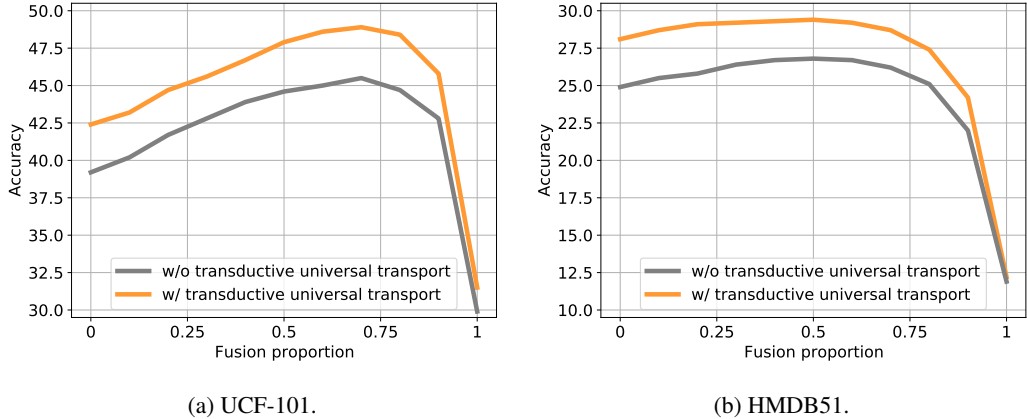

(a) UCF-101.

(b) HMDB51.

Figure 3: **Fusing action and object information** for zero-shot recognition on UCF-101 and HMDB51. Combining universal action and object information benefits zero-shot recognition, with transductive universal transport preferred regardless of the fusion proportion between both sources.

test video, provides a boost to 31.6% with the top 1,000 objects. We find that as long as the threshold is not set too strictly (*e.g.,* keeping 1,000 objects or more) provides stable zero-shot results.

In Table 1 on the right, we show that the proposed weighting matters. For this Table we keep the top 1,000 objects and investigate all four combinations of uniform and proposed weighting. With uniform weights for both unseen actions and objects, the results are similar to the baseline object-based setup, while the results improve when incorporating either or both of the weight vectors to the proposed setup. We conclude that also in the universal object-based model for zero-shot action recognition, transductive universal transport is beneficial.

**Fusion.** With our approach shown to be effective for both universal action and object approaches separately, we investigate the potential of fusing both perspectives. In Figure 3, we show the effect of fusing both approaches with and without transductive universal transport on UCF-101 and HMDB51. On UCF-101, we find that combining both approaches has a clear positive effect. Our results improve from 42.4% (universal actions) and 31.5% (universal objects) to 47.9% when balancing both equally. When setting the fusion proportion to 0.3, the results can even be further improved to 48.9. Due to the zero-shot nature of our approach, we stick to an *a priori* equal balance between both setups. Averaged over all fusion proportions, our approach provides a boost of 3.0 percent point compared to the baseline fusion.

In Figure 3, we also show the fusion results for HMDB51. Our approach improves the accuracy from 24.9% to 28.1% when using universal action models only. The results are further improved to 29.4% with an equal fusion of our universal object representations. We conclude that universal seen actions and objects are complementary for zero-shot action recognition and their fusion benefits from transductive universal transport.

| | UCF-101 | | | HMDB51 | | |
|---|---|---|---|---|---|---|
| | 20 | 50 | 101 | 10 | 25 | 51 |
| Hahn et al. (2019) | $36.5 \pm -$ | $22.1 \pm -$ | - | $40.1 \pm -$ | $23.5 \pm -$ | - |
| Bishay et al. (2019) | $42.7 \pm 5.4$ | $23.2 \pm 2.9$ | - | - | $19.5 \pm 4.2$ | - |
| Mishra et al. (2020) | - | $26.1 \pm 3.0$ | - | - | $21.3 \pm 3.2$ | - |
| Mandal et al. (2019) | - | $38.3 \pm 3.0$ | - | - | $30.2 \pm 2.7$ | - |
| Mettes & Snoek (2017) | $51.2 \pm 5.0$ | $40.4 \pm 1.0$ | 32.8 | - | - | - |
| Zhu et al. (2018) | $53.8 \pm -$ | $42.5 \pm -$ | 34.2 | - | - | - |
| Mettes et al. (2021) | $61.1 \pm -$ | $47.3 \pm -$ | 36.3 | - | - | - |
| Kim et al. (2021) | - | $48.9 \pm 5.8$ | - | - | - | - |
| Brattoli et al. (2020) | - | $49.2 \pm -$ | 39.8 | - | $32.7 \pm -$ | 26.9 |
| **This paper** | **64.1** $\pm 10.5$ | **54.4** $\pm 3.2$ | **47.9** | **47.1** $\pm 8.8$ | **36.0** $\pm 5.6$ | **29.4** |

Table 2: **State-of-the-art comparison on UCF-101 and HMDB51** for different numbers of test actions. Across both datasets and different numbers of test actions, our approach obtains the highest zero-shot action classification performance.

| | UCF Sports | | | | | J-HMDB | | | | |
|---|---|---|---|---|---|---|---|---|---|---|
| | 0.1 | 0.2 | 0.3 | 0.4 | 0.5 | 0.1 | 0.2 | 0.3 | 0.4 | 0.5 |
| Jain et al. (2015) | 38.8 | 23.2 | 16.2 | 9.9 | 7.2 | - | - | - | - | - |
| Mettes & Snoek (2017) | 43.5 | 39.3 | 37.1 | 35.7 | 31.1 | 34.6 | 33.3 | 30.5 | 26.8 | 23.0 |
| Mettes et al. (2021) | 47.3 | 43.0 | 40.7 | 37.9 | 33.1 | 37.3 | 37.1 | 33.9 | 31.0 | 26.7 |
| **This paper** | **51.1** | **47.8** | **45.7** | **41.2** | **33.5** | **44.2** | **43.5** | **40.1** | **35.5** | **30.8** |

Table 4: **State-of-the-art comparison on UCF Sports and J-HMDB** for five overlap thresholds. Across datasets and overlap thresholds, we obtain the highest scores, highlighting the effectiveness of our approach in the context of zero-shot spatio-temporal localization.

## 4.3 STATE-OF-THE-ART COMPARISON

**Action recognition.** In Table 2, we compare our results on UCF-101 and HMDB51 to the state-of-the-art in zero-shot action recognition. Similar to other universal approaches, in the scenarios with random sub-selection of the test actions (20 and 50 for UCF-101, 10 and 25 for HMDB51) we do not use the remaining actions and their videos for network training. On both datasets, the current state-of-the-art is given by Brattoli et al. (2020). On UCF-101 with 101 test actions, they obtain an accuracy of 39.8%. With our transductive universal transport applied and fused over universal action and object models, the results improve to 47.9%. We also improve over the recent universal object model of Mettes et al. (2021) and the action signature approach of Kim et al. (2021) across all numbers of test actions. On HMDB51, we improve the state-of-the-art with 2.5 p.p. (51 actions), 3.3 p.p. (25 actions), and 7.0 p.p. (10 actions). We conclude that transductive universal transport is a viable solution for universal action and object models in zero-shot action recognition. Lastly, we also include a one-to-one comparison to universal action and object models in Table 3 for fair comparisons.

| | Baseline | + This paper |
|---|---|---|
| Universal action model $\star$ | 39.2 | **42.4** |
| Universal object model $\dagger$ | 29.9 | **31.6** |

$\star$ pre-trained model and word embeddings from (Brattoli et al., 2020)
$\dagger$ pre-trained object model and action-scoring given by Mettes et al. (2021), word embeddings from (Brattoli et al., 2020)

Table 3: **One-to-one overview** of our positive impact on universal action and object models.

**Spatio-temporal action localization.** Lastly, we showcase the potential of our approach for improving zero-shot spatio-temporal action localization as first introduced by Jain et al. (2015). Since our approach operates over entire videos, we start from the spatio-temporal tubes made publicly available by Mettes et al. (2021). For each tube, we simply add the score for each action from the entire video as given by our approach. In Table 4, we report the AUC scores for UCF Sports and J-HMDB. Across datasets and overlap thresholds, we find that the global scores from our approach boosts spatio-temporal localization. This is because the scores of our approach help to distinguish and rank tubes from different videos as they encode contextual information.

## 5 RELATED WORK

Zero-shot action recognition refers to the task of assigning an action label to a test video given a pool of action labels not observed during training. A common approach is learn and transfer a shared representation on seen actions with training examples to be able to perform inference on unseen actions. A well-known shared space is given by attributes (Liu et al., 2011; Zhang et al., 2015; Gan et al., 2016b). By projecting test videos to an attribute space, inference is possible through a neighbour search with unseen actions manually defined in the same space. Since attributes require manual annotations for every action, more recent works prefer to use word embeddings (Gan et al., 2016a;c; Li et al., 2016; Bishay et al., 2019) or action hierarchies (Long et al., 2020) to provide a share space for seen and unseen actions.

State-of-the-art approaches to zero-shot action recognition take a universal learning perspective towards the use of semantic word embeddings as the shared space for knowledge transfer. Rather than relying on a small set of seen actions from the same dataset, large-scale models are trained on hundreds or thousands of seen categories to learn a direct mapping from videos to the shared space. The first universal learning direction relies on large-scale actions with training videos. Zhu et al. (2018) were the first to propose a large-scale universal action perspective by learning a video network on 200 actions from ActivityNet (Caba Heilbron et al., 2015). Recently, Brattoli et al. (2020) have obtained the best performance in zero-shot action recognition by scaling this perspective to 664 actions from Kinetics (Carreira & Zisserman, 2017). Due to the large amount of seen actions, care needs to be taken to avoid (near) duplicates between the seen and unseen actions, as also noted by Roitberg et al. (2018).

Next to universal action models, several works have shown competitive results by taking a universal object perspective. Large-scale networks are trained in the image domain on thousands of object labels (Jain et al., 2015; Wu et al., 2016; Mettes & Snoek, 2017; Liu et al., 2019; Mettes et al., 2021). Once trained any action can be inferred based on the object likelihoods in test video and the semantic similarities between objects and unseen actions. Regardless of the universal perspective, the current state-of-the-art shares the same fate: zero-shot inference results in a long-tailed distribution over selected actions. Many unseen actions are simply never selected during persistent biases. We tackle this problem with the transductive universal transport over universally trained models. This paper introduces a new way of dealing with biases in universal models, hence the focus on the video domain over the image domain, where universal models are not state-of-the-art.

Within zero-shot action recognition, a transductive view is commonly applied, as noted in (Estevam et al., 2021). Rohrbach et al. (2013) provide a foundation for transduction in zero-shot context by exploiting the inter-sample similarity over the test set. In similar spirit, several works have proposed transductive extensions for zero-shot action recognition (Fu et al., 2014; Kodirov et al., 2015; Alexiou et al., 2016; Xu et al., 2017; Gao et al., 2019; Mandal et al., 2019). While transductive learning no longer operates on individual test videos, many applications require evaluating entire video batches/collections simultaneously. Different from existing approaches, we use the distribution over all test videos to improve the unseen action embeddings in the shared output space of universal models by building on optimal transport. Moreover, our approach can naturally switch between inductive and transductive settings, as only the positioning of action embeddings are updated.

## 6 CONCLUSIONS

In this work, we investigate a persistent limitation in current universal learning models for zero-shot action recognition, namely selection biases in the assignment of unseen actions to test videos. We introduce transductive universal transport to alleviate this bias. Our approach consists of three stages: (i) finding an optimal transport mapping from unseen action embeddings to the mapped test videos (in universal action models) or object embeddings (in universal object models), (ii) calculating a target embedding for each unseen action as a weighted Fréchet mean with the weights given by the transport couplings, and (iii) re-positioning the unseen actions along the geodesic spanned by the original and target embeddings. Empirical evaluation on four datasets shows the effectiveness of our approach for debiasing action assignments and improving zero-shot recognition as a result, outperforming existing approaches for zero-shot action classification and zero-shot spatio-temporal action localization. Our approach is general and can be used to improve any universal model.

## ETHICS AND REPRODUCIBILITY STATEMENT

Action recognition has a broad range of applications, for example in autonomous driving and robotics. While such applications hopefully have a positive societal effect, the research can also potentially be used for applications such as surveillance and this should be taken into account. To ensure reproducibility, we start from off-the-shelf models as provided by current universal learning models. In the paper, we have outlined the hyperparameters of our approach and we will release the code to reproduce the experiments.

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
