# OpenReview forum: "Transductive Universal Transport for Zero-Shot Action Recognition"
_ICLR.cc/2022/Conference — ICLR 2022 Submitted_

### Official Review · Reviewer_Gn5W · 2021-11-01

**Correctness:** 3
**Technical Novelty And Significance:** 3
**Empirical Novelty And Significance:** 3
**Recommendation:** 5
**Confidence:** 2

**Main Review:**

Strengths
1. The use of transductive universal transport for zero-shot action recognition is new.
2. The experimental results show the effectiveness of this new method and also better performance than prior states of the art.

Weaknesses
1. The approach needs access to the entire testing video set to obtain distribution information of testing videos. This is an unrealistic setting. When used in practice, a machine learning model should expect to see one testing example a time.

2. Many symbols are not clearly defined, making the math descriptions in this paper hard to read. For example, in Eq. (2), it is unclear what w_s, w_u, sum_|Lu| sum_|Ls| are, what weights for labels, i.e., w_lu and w_ls, mean, and why u_s and u_u are sets of labels.

3. Does the use of transductive universal transport bring any computational overhead to the zero-shot learning model? It is interesting to see comparison of inference time and complexity with the baseline and some states of the art.

**Summary Of The Paper:**

This work introduces transductive universal transport for zero-shot action recognition, where no training examples for unseen classes are available. To address the biases of prior approaches towards seen classes during inference, this paper re-positions unseen action embeddings through transduction by using the distribution of the unlabelled test set. Experimental results on several action recognition datasets demonstrate the effectiveness of the proposed method.

**Summary Of The Review:**

The idea of using transductive universal transport for zero-shot action recognition is new, and the performance is good. But the core setting that the entire testing set is available during training to get the distribution information is unrealistic. The writing, especially the math part, needs improvement.

---

> ### Author Response · Authors · 2021-11-14
> **Rebuttal**
>
> We thank the reviewer for their positive comments on the novelty and effectiveness of the proposed approach. Below, we have addressed the raised concerns:
>
> **Practicality of transductive action recognition**
>
> We agree that certain applications require evaluating one testing example at a time. In practice however, many applications seek to perform inference on entire batches/collections of videos, including but not limited video recommendation, search in video datasets, and tagging video collections. All such applications benefit from the proposed approach. We also note that our approach can effortlessly switch between inductive and transductive settings, as we only change the locations of the action embeddings based on transduction. I.e. our approach does not break down if the transductive assumption fails. We have clarified the practicality of transductive action recognition and the robustness of our approach with respect to the transductive assumption in Section 5.
>
> **Symbol definitions**
>
> Following the reviewer’s suggestion, we have added the definitions of the symbols used in Equations 2 and 3 below the equation. Thank you.
>
> **Computational overhead**
>
> Compared to the universal action baseline, our approach requires additional transport and weighted Frechet mean calculations. Given an offline computed clustering, this calculation is 0.3 seconds per action. With the computed target embeddings, the inference time is equal to the universal action baseline.
>
> We hope to have alleviated any concerns regarding practical applicability and mathematical clarity of the paper.

---

### Official Review · Reviewer_EmnY · 2021-11-03

**Correctness:** 3
**Technical Novelty And Significance:** 2
**Empirical Novelty And Significance:** 2
**Recommendation:** 5
**Confidence:** 4

**Main Review:**

Strengths

- Work on the problem of reducing the biases between seen categories in the source domain and unseen categories in the target domain
the semantic space.

- Evaluate the approach on benchmark datasets for two tasks:  zero-shot action classification and spatio-temporal action localization.

Weaknesses

- Novelty seems incremental. The proposed transductive universal transport algorithm for embedding reposition seems like a simple weighting method guided by the distribution of the unlabelled test set. The paper merely uses existing approaches to solve the transductive optimal transport problem but does NOT bring any new insights.

- Generalization seems a concern. The proposed approach heavily depends on the distribution of the unlabelled test set. It seems sensitive to the distribution and the number of clusters.  As shown in Figure 2, using the target embeddings seems on par with repositioned embeddings. Also, the proposed approach seems to only works in the case with a small number of clusters.

**Summary Of The Paper:**

This work tries to address the problem of zero-shot action recognition. Particularly, the paper aims at preventing the case that many unseen action categories in the target domain are simply never being selected during inference. Using the distribution of the unlabelled test set, the embeddings of unseen actions in the target domain are reweighted and repositioned along the geodesic such that they are better aligned with embeddings of training actions in the source domain. In experiments, Empirically, the proposed method has been evaluated on
benchmark datasets for tasks zero-shot action classification and spatio-temporal action localization.

**Summary Of The Review:**

This paper proposes a sensible solution to reduce the bias between the source domain and the target domain for the task of action recognition. But novelty seems incremental. Also, some experiments seems a bit unconvincing and the approach seems not to scale to general settings.

---

> ### Author Response · Authors · 2021-11-14
> **Rebuttal**
>
> We thank the reviewer for their positive words on the proposed solution and the opportunity to better emphasize the new insights and generalization capabilities that come with the paper.
>
> **New insights in zero-shot action recognition**
>
> We apologize for not making the new insights of this paper for zero-shot action recognition more explicit. This paper provides the following new insights:
> 1. We identify a fundamental limitation in universal models for zero-shot action recognition, namely strong biases towards subsets of actions, resulting in many actions that are simply never selected due to a mismatch between embedded videos and unseen actions.
> 2. We introduce Transductive Universal Transport for zero-shot action recognition. Our approach extends optimal transport in two ways: (i) we propose to obtain per-class target embeddings through a Fréchet mean over all video clusters, weighted by their optimal transport assignment; (ii) we introduce a semantic regularization over target embeddings, defined as a spherical interpolation between the original and target embeddings per action.
> 3. Our approach is general and improves both universal action and object models. Due to the general nature of our approach, we are also able to integrate our approach in zero-shot spatio-temporal action localization. Overall, we obtain state-of-the-art results for both zero-shot classification and localization on four action datasets.
>
> We have made the new insights of our approach more clear in the introduction of the paper.
>
> **Generalization**
>
> We have improved the visualization in Figure 2a to address the two questions regarding generalization. The new Figure 2a better visualizes that repositioning does improve results compared to using the original embedding. The best results are obtained by interpolating between the original and target embeddings. The figure also addresses the second concern: our approach is robust to the number of clusters.. When using 1,000 clusters or more, we find stable and improved results over using the original embeddings.
>
> We thank the reviewer for their guidance and hope to have addressed all concerns.

---

### Official Review · Reviewer_2kAP · 2021-11-03

**Correctness:** 3
**Technical Novelty And Significance:** 2
**Empirical Novelty And Significance:** 2
**Recommendation:** 5
**Confidence:** 2

**Main Review:**

While the paper demonstrates state-of-the-art results, one important concern is about fairer comparisons, especially the zero-shot spatio-temporal localization experiments (Table 3).
- The setting of the proposed method is different from some compared papers.  For example, the authors focus on transductive ZSL, while Mettes et al. (2021), Kim et al. (2021),  and Brattoli et al. (2020) focus on inductive ZSL.
- The proposed method uses both action and object information, while Brattoli et al. (2020) use action information only, and Mettes et al. (2021) use object information only.

Without fairer comparisons, it is hard to assess the effectiveness of the proposed method.

Another concern is that the importance of some critical components is not adequately evaluated.
- This is also related to the comments above mentioned. The proposed method uses both video features and object information. Is this critical to obtain a good performance? The importance of video features and object information is not properly evaluated. One way to show this is to evaluate the performance of the proposed method using only one type of modality. Partial information is given in Figure 3 and the Fusion paragraph on page 7. Based on Figure 3 and the discussion, it seems that the proposed method does not outperform Brattoli et al. (2020) and Mettes et al. (2021) under the same experimental settings as the compared methods?

Typo: in 3.2 implementation details: (2+1)D -> R(2+1)D


**Summary Of The Paper:**

The paper targets transductive zero-shot action recognition.  To alleviate models biased to seen categories, the authors propose to re-position unseen action embedding through transduction. There are three steps in the proposed method: first, finding an optimal mapping from unseen actions to the mapped video in the shared hyperspherical space. Second, defining target embeddings as weighted Frechet means with the weight given by the transport couplings. Third, re-positioning unseen action embeddings along the geodesic between the original and target. The zero-shot classification performance of the proposed method is tested on the UCF-101 and HMDB datasets. The zero-shot spatio-temporal localization performance is tested on the UCF Sport and J-HMDB datasets.


**Summary Of The Review:**

The paper can be further strengthened by demonstrating fairer comparisons and adequately evaluating the importance of the critical components.

---

> ### Author Response · Authors · 2021-11-14
> **Rebuttal**
>
> We thank the reviewer for their suggestions to make comparisons more explicit and direct, as well as evaluating critical components.
>
> **Direct comparison to action-only and object-only models**
>
> Following the reviewer’s suggestion, we have included a table to the paper that provides a more explicit comparison to action-only and object-only universal models. The table shows that our approach directly improves zero-shot recognition results, while using the exact same models, action scoring function, and word embeddings.
>
> | | Baseline | + This paper |
> |-|-|-|
> | Universal action model \star | 39.2 | 42.4 |
> | Universal object model \dagger | 29.9 | 31.6 |
>
> \star = pre-trained action model and word embeddings given by Brattoli et al. (2020)
>
> \dagger = pre-trained object model and action-scoring given by Mettes et al. (2021), word embeddings given by Brattoli et al. (2020)
>
> For universal action models, our performance improves over both the model provided by the authors and the state-of-the-art numbers as given in (Brattoli et al., 2020). For Mettes et al. (2021), we improve over their base model and note that their final performance is indeed higher. This is because their approach employs additional tricks to boost zero-shot recognition results, including better word models and incorporating prior knowledge about object discrimination, object language, and object naming. Overall, we conclude that our approach is a general solution for both universal action and object models. We have added the direct comparison to the paper and we have fixed the typo. Thank you.

---

### Official Review · Reviewer_cLxn · 2021-11-04

**Correctness:** 3
**Technical Novelty And Significance:** 3
**Empirical Novelty And Significance:** 3
**Recommendation:** 6
**Confidence:** 2

**Main Review:**

Strength:
1. The zero-shot learning for action recognition is good direction however the applicability is limited in real world scenario.
2. Figure 1 is very clear to give the illustration of the concept and the high-level idea of the whole paper.
3. The performance are very stronger when compared with other baselines.

Weakness:
1. The technical details is a little hard to follow since I am not very familiar with the zero-shot learning. In my opinion, the proposed method is a general approach across video and image domains. Why not implement the experiments of such method on the image zero-shot learning?
2.  Is there any visualization to intuitively show the learning results by the transductive universal transport?

**Summary Of The Paper:**

The paper presents a zero-shot action recognition framework by learning an universal mapping from video to semantic space. The unseen action embeddings are re-positioned through leveraging the distribution of unlabelled test set. The universal mappings from unseen action to test videos are first defined and the target embeddings are treated as weighted Frechet means. The unseen action embeddings are re-positioned as a semantic regularization. The results on UCF101 and HMDB-51, UCF Sports and J-HMDB validates the proposed method.

**Summary Of The Review:**

The paper gives a new method for zero-shot action recognition by learning the transduction from unseen action to test videos in hyperspherical space. The performances are good when compared with other state-of-the-art methods. However, the details of the technique  is a little hard to follow for me. In my opinion, the paper forms well and the writing is very good. I give accept currently because of my unfamiliarity of such domain. By the way, I will follow up the work in the following reviewing sections.

---

> ### Author Response · Authors · 2021-11-14
> **Rebuttal**
>
> We thank the reviewer for their positive words about paper clarity and its strong performance over baselines. Below we have addressed the raised questions:
>
> **Why focus on videos?**
>
> We agree with the reviewer that the approach can generalize to the image domain. Due to the lower-dimensional nature of the data, the image domain however offers more possibilities to overcome biases in zero-shot recognition, e.g. feature or image generation for unseen classes. Such approaches have not found success in the video domain, where in contrast universal models have been most successful. This paper seeks to overcome inherent limitations in universal models, hence our focus on the video domain. We have included this motivation to the related work.
>
> **Visualization**
>
> Following the reviewer’s suggestion, we have made a visualization of the transductive universal transport when applied on the universal action model of Brattoli et al. (2020). The figure shows a t-SNE plot of the embedding of videos and actions in UCF-101 before and after transductive universal transport. The figure shows how some action embeddings are a priori isolated and never considered as target for any action. After the proposed transport, the alignment between test videos and unseen actions is recovered, leading to a more balanced action selection and improved zero-shot recognition performance. We will add the visualization to the supplementary materials, due to lack of space in the main paper.
>
> We thank the reviewer for their guidance and hope to have addressed their concerns.

---

### Decision · Program_Chairs · 2022-01-20

**Decision:**

Reject

**Comment:**

This paper was reviewed by four experts in the field and received mixed scores (1 borderline accept, 3 borderline reject). The reviewers raised their concerns on lack of novelty, unconvincing experiment, and the presentation of this paper. AC feels that this work has great potential, but needs more work to better clarify the contribution and include additional ablated study. The authors are encouraged to consider the reviewers' comments when revising the paper for submission elsewhere.